# Increasing seasonal variation in the extent of rivers and lakes from 1984 to 2022

Björn Nyberg[1,2,3*], Roger Sayre[4], Elco Luijendijk[1]

Department of Earth Sciences, University of Bergen, Allegaten 41, 5020, Bergen, Norway.[1]

Bjerknes Centre for Climate Research, Allegaten 70, 5020, Bergen, Norway.[2]

7Analytics, Innovation District Solheimsviken 7c, 5054, Bergen, Norway[3] U.S.

Geological Survey, 516 National Center, Reston, VA, 20192, USA.[4]

 Correspondence: bn@7analytics.no

**Abstract**

Knowledge of the spatial and temporal distribution of surface water is important for water resource management, flood risk assessment, monitoring ecosystem health, constraining estimates of biogeochemical cycles and understanding our climate. While global scale spatial-temporal change detection of surface water has significantly improved in recent years with planetary scale remote sensing and computing, it has remained challenging to distinguish the changing characteristics of rivers and lakes. Here we analyze the spatial extent of permanent and seasonal rivers and lakes globally over the past 38-years based on new data of river system extents and surface water trends. Results show that while the total permanent surface area of both rivers and lakes has remained relatively constant, the area with intermittent seasonal coverage has increased by 12% and 27% for rivers and lakes, respectively. The increase is statistically significant in over 84% of global water catchments based on Spearman rank correlations (rho) above 0.05 and p values less than 0.05. The seasonal river extent is nearly 32% larger than previously observed annual mean river extent, suggesting large seasonal variations that impact not only ecosystem health but also estimations of terrestrial biogeochemical cycles of carbon. The outcomes of our analysis are shared as the Surface Area of Rivers and Lakes (SARL) database, serving as a valuable resource for monitoring and research of hydrological cycles, ecosystem accounting and water management.

## 1. Introduction

Climate change and population growth have placed considerable stress on our natural freshwater resources. Water demand has increased nearly 8-fold over the past century with an estimated 70% of the total used to meet irrigation needs (Siebert et al., 2010; Wada et al., 2016). To meet the increasing water demand, an estimated 16.7 million reservoirs have been built (Lehner et al., 2016) with the largest 24783 dams holding a predicted 7384 $km^3$ of freshwater (Wang et al., 2023). Current water demand represents only 10% of our approximate annual renewable freshwater resources (Oki and Kanae, 2006). Nonetheless, water scarcity remains a significant problem around the

world due to the variability of water in time and space (Oki and Kanae, 2006; Mekonnen et al., 2016). The hydrological cycle is also crucial to the health of our ecosystems and biodiversity that depend on the recurrence and seasonality of water to support life (Gleeson et al., 2020). Furthermore, inland waters are also an important component in biogeochemical cycles of $CO_2$ and methane that, by size, disproportionately contribute a significant portion to our total greenhouse emissions annually (Bastviken et al., 2004; Allen and Pavelsky, 2018; Matthews et al., 2020).

While the surface area of water is only one part of the hydrological cycle, it is the most accessible portion influencing human and ecosystem behavior and an important component in groundwater recharge (Oki and Kanae, 2006; Sibert et al., 2010; Gleeson et al., 2020). Knowledge of the type of waterbody, i.e. whether it is permanent, intermittent, or seasonal and whether it is part of a river system or a lake, is important to understand the role of water bodies in different hydrological processes, ecosystem support and biogeochemical cycles. The changing physical environment and its waterbody type due to droughts, floods or direct human alteration also alters migration patterns of humans, ecosystems and biodiversity (Neumann et al., 2015; Van Loon et al., 2016). In addition, the perennial and seasonal state of both rivers and lakes has important implications for ecosystem health (Messager et al., 2021) and carbon cycles (Keller et al., 2020). The type, extent and seasonality of waterbodies at a global basin scale is needed for improved water resource management and sustained delivery of ecosystem services (Sheffield et al., 2018).

Planetary scale computing and analysis of remotely sensed imagery have led to a number of studies revealing the unprecedented impact of human resource management and climate change stress on the extent of water (Van Dijk et al., 2011; Wada et al., 2016; Pekel et al., 2016, Donchyts, et al., 2016). In particular, significant advancements have been made in identifying and quantifying the historical change in global reservoirs at a 0.01 to 100 $km^2$ resolution (Donchyts et al., 2022). In addition, the temporal analysis of lakes extents has been analyzed up to 50 degrees North at a 0.1 $km^2$ resolution (Khandelwal et al., 2022). The identification of natural lakes in high latitude regions has not been analyzed within a global water surface change context.

Far fewer studies have analyzed the change of river extents. Allen and Pavelsky (2018) mapped the observed global surface area of rivers but only for a specific year and at mean annual water discharge. More recently, Feng et al., (2022) quantified the temporal variability in global river widths over the past 37 years based on 30 m Landsat imagery. However, this study does not map the changing surface area of rivers nor provide measurements at the confluence and divergence of rivers common in anabranching and braided systems that comprise an estimated 52% of global rivers (Nyberg et al., 2023). As a result, there remains a significant knowledge gap in the temporal variability of river surface area and its interaction with lakes and reservoirs. The aim of this paper is to assess the utility of a new, global river extent dataset (Nyberg et al., 2023) in mapping the historical change in water surface area for rivers and lakes over the past 38-years and to examine the implications for water resource management, ecosystem health and biogeochemical cycles.

## 2. Materials and Methods

### 2.1 Water Surface Area Classification

To classify the permanent and seasonal extent of lakes and rivers, we utilize existing datasets describing the surface area of water combined with a new and improved definition of river extents on a global scale. Our analysis is based on the Global Surface Water (GSW) version 1.4 by Pekel et al. (2016) that describes the permanent and seasonal extent of open water from 1984 to 2022 based on 30 m Landsat imagery. Permanent surface area of water is defined as locations where open water is detected for all twelve months of a given year (or 100% of valid pixels). Seasonal surface water was defined as any pixel location with at least one month during which water was detected. The authors report less than 1% false positive open water classifications and less than 5% missed open water classifications based on 40,000 randomly selected points.

To define the spatial extent of rivers is challenging given the dynamic nature of rivers, varying morphology and perennial versus non-perennial character of rivers. We used a dataset produced by Nyberg et al. (2023) that quantified the global extent of river channel belts (GCBs) at a 30 m Landsat imagery resolution using a machine learning method that spatially delineates channel belt areas based on geomorphological features. In this dataset, the river channel belt extents show the riverine landforms of the river channel and its associated levees, bars and overbank deposits that therefore capture the evolution of the riverine environment over time (Figure 1). The model reports a confidence value ranging from 0 to 100%, where a 0% confidence value indicates a non-riverine environment whereas a 100% confidence indicates a riverine environment for a given pixel location. The model used by Nyberg et al. (2023) reports a 94% accuracy to the validation dataset for channel belts wider than a 1 km.

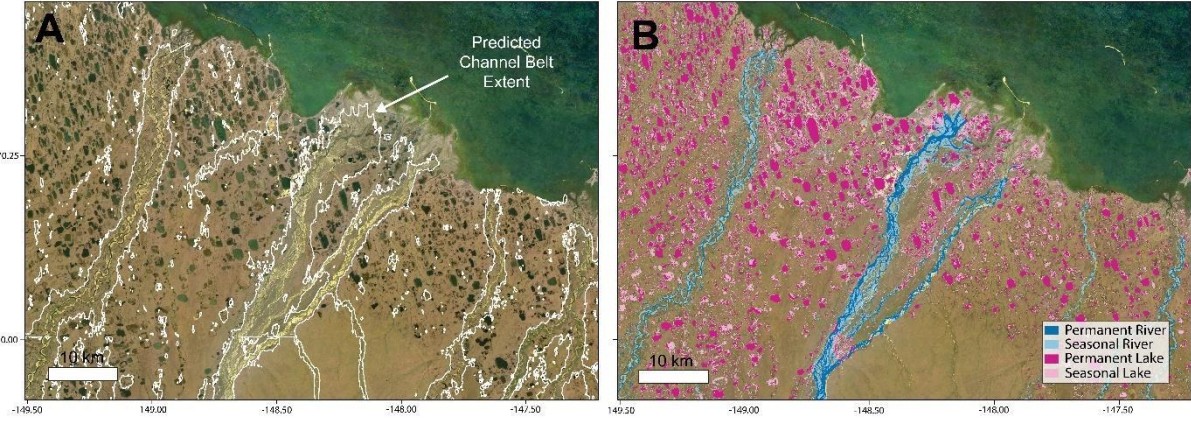

Figure 1: Example permanent and seasonal water extent in rivers and lakes - A) Example Landsat 8 imagery for 2020 with overlain delineations of the maximum channel belt extent in white based on the GCB dataset (Nyberg et al., 2023). Any pixel outside the channel belt is defined as lacustrine/wetland or floodplain. B) Permanent and seasonal extent of rivers and lakes for the year 2020 based on the new Surface Area of Rivers and Lakes (SARL) database. Landsat 8 imagery courtesy of the US Geological Survey.

To improve delineations of river channel belt extent we refine the classification of Nyberg et al. (2023) by using
GCB pixels with a reported confidence of 10% or higher and a 50% confidence above 60 degrees North. This step
was chosen to constrain rivers without distinct channel belts in high latitude regions (e.g., Canadian Shield or
Siberian Plateau) given most sedimentary basins with clearly defined channel belts occur around mid to low
latitude regions (Nyberg and Howell, 2016). In addition, previous databases of large lakes and reservoirs (~ > 10
km$^2$) defined by the HydroLakes (Messager et al., 2016) and OpenStreetMap (2022) datasets are included to further
improve delineations. This step was achieved by converting the vector lacustrine databases to rasters at the same
30 m resolution of the GCB delineation to remove the misclassified pixels. The inclusion of the lacustrine databases
reduced the global channel belt extent by 1.86% or 13.4 x 10$^4$ km$^2$.
Finally, the seaward extent of the SARL database is based on the Global Shoreline Vector (GSV) dataset by Sayre
et al. (2019). This classification represents an image-derived instantaneous shoreline position for the year 2014
capturing between a low- and high-tide classification. This step is necessary to remove classification of sea pixels
from the resulting lacustrine/riverine classification. Depending on the tidal range for a particular region and
available Landsat images, this may result in a lower or higher riverine and lacustrine extent, which may impact
our subsequent statistical analyses. However, the GSV dataset provides a global, 30 m resolution shoreline
classification, creating a consistent definition of the shoreline that is needed for the SARL database.
By combining the GCB and GSW datasets, we produce a new global dataset mapping the historical change of the
seasonal and permanent water surface area of lakes and rivers (SARL) from 1984 to 2022. The seasonal extent of
water within the channel belt shows rivers at bankfull or larger flood events with inundation persisting for at least
one month. Water Bodies outside the channel belt are defined as a lakes or wetland regions (Figure 1). The
processing is completed on the Google Earth Engine platform (Gorelick et al., 2017) resulting in a global database
of the seasonal and permanent surface area of rivers and lakes from 1984 to 2022 at a 30 m Landsat resolution.

**2.2 Temporal Water Surface Area Analysis**

Following the mapping of the SARL database, we analyze the data by aggregating results on drainage catchments
derived from the HydroSHEDS level 5 catchment dataset (Lehner et al., 2008). Considering that satellite imagery
is not available for certain years due to non-acquisition or excessive cloud cover, it is important to consider missing
values in the time series of surface water observations. Pekel et al., 2016 define the location of missing values for
each year in the GSW database but do not identify the waterbody type or seasonality of those missing values. To
rectify this omission, we take the averaged ratio between seasonal to permanent waterbody extent and waterbody
type (lake, river and no water) ratio between 2015 to 2017 for each catchment as a baseline given there are no
reported missing values during those years. While this approach does not account for the changing seasonal to
permanent water ratios or waterbody type over time, it does provide an approximation that allows for the analysis
of water trends in situations where a few missing values (< 5%) are reported in a much larger catchment region.
This method was preferred over a long-term pixel average and interpolation given that earlier landsat acquisitions
have a lower temporal resolution, and therefore seasonal changes can be interpreted less reliably, which may result
in skewed ratios of seasonal to permanent rivers and lakes.

Subsequently, for each catchment the permanent and seasonal water extent for each time period without satellite
data is calculated proportional to the number of missing values and known ratio of seasonal to permanent extent
in equation 1.

$$pArea = pO + (nD * (1 - k)) \quad if\ nD * (1\text{-}k) < 0.05 \qquad (1a)$$

$$sArea = sO + (nD * k) \quad if\ nD * k < 0.05 \qquad\qquad (1b)$$

where pArea is the permanent surface area, sArea is the seasonal surface area, pO is the observed permanent
surface area, sO is the observed seasonal surface area, nD is the area of no data values for the entire catchment, k
is the seasonal:permanent surface area ratio and p is the total area of water pixels (e.g. observed + missing values).

Equation 1 is processed separately for rivers and lakes and only assigned to catchments where less than 5% of the
data is missing for any given year, waterbody type and seasonality. For years with more than 5% missing values,
the first year with valid observations for any given catchment is used. This creates an accompanying dataset
showing the absolute change, percentage change and annual percentage change in surface area of water bodies for
each catchment from 1984 to 2022 (see Data Statement section for interactive map and supplementary figure S1).

To statistically analyze trends in the surface area of rivers and lakes by catchment, we perform Spearman rank
correlations (Spearman, 1987). This correlation measures the monotonicity of the relationship between two
parameters, in this case the surface area of the waterbody extent versus time. This measure is ideal considering
waterbody extent is often not a linear relationship with time given the interannual variability. To perform this
analysis, we limit our analyses to catchments with more than 10 years of results, at least $1km^2$ of detected water
surface area and 95% data coverage for each year. Statistically significant trends are defined by p value less than
0.05 or 5%. In total, the analysis captures between 1172 to 1493 watersheds (or 34 to 41% of the global watersheds
with water occurrence), depending on the seasonal versus permanent and river versus lake analysis performed.

**2.3 Validation**

To validate and assess the accuracy of the SARL dataset over the 38-years of available water surface data, we
compare the results to the manually interpreted extent of river channel belt and lake environments for the year
2022. In total, 50 locations (see supplementary material Figure S2) measuring 50 $km^2$ were randomly selected
(excluding Greenland and Antarctica) to manually map the river channel belt extent using 30 m Landsat 8 imagery.
The Landsat imagery corresponds to the same spatial resolution and year of acquisition as the GCB dataset defined
by Nyberg et al. (2023) which is used in the current model to define the global channel belt extent (section 2.1).
The manual delineation of the channel belt is defined as the encompassing region of the active river and its
associated bars, over bank deposits and abandoned channels (Nyberg et al., 2023) to show the maximum
geomorphologically-observed extent of the river system through time.
Following the proposed automated method to classify the current SARL database (e.g., Figure 1), wate rbodies
defined within the manually defined 2020 channel belt extent are classified as riverine whereas all other surface
water bodies are defined as lacustrine. Seasonal and permanent waterbody extent for both rivers and lakes are then
extracted based on the GSW model (Pekel et al., 2016). The manual river channel belt delineation is subsequently
used to compare the accuracy of the automated delineation in capturing the permanent and seasonal water extent
of the GSW dataset from 1984 to 2022 (see supplementary Table S2). While the GCB dataset and manual
delineation for validation is only available for the year 2022, the channel belt represents the lateral migration of a
rivers course through time (Nyberg et al., 2023), thus capturing the extent of the river system through multiple
years. Hence, the method provides a baseline assessment for the accuracy of the automated method in capturing
the natural variability in seasonal and permanent water extent through time.

**3. Results**

**3.1 Total Water Surface Extent**
The permanent surface area of rivers has remained relatively steady over the past 38 years, increasing slightly by
1.1% to a total area of 2.9 x $10^5$ km$^2$ (Figure 2). In contrast, the observed seasonal extent of rivers has increased
more significantly by 12% with a total area of 3.2 x $10^5$ km$^2$ by 2022. The yearly percentage of seasonal to
permanent river water extent ranges from 88 to 119%, increasing towards the end of 2022. Similarly, the spatial
extent of permanent lake surface area has increased by less than 1% since 1984 to a total area of 24.2 x $10^5$ km$^2$
(Figure 2). The seasonal extent of lakes has however increased significantly, by as much as 27% since 1984 to a
total area of 7.2 x $10^5$ km$^2$. The ratio of seasonal to permanent water extent in lakes is considerably lower than that
of rivers and ranges between 23-31% over the same period.

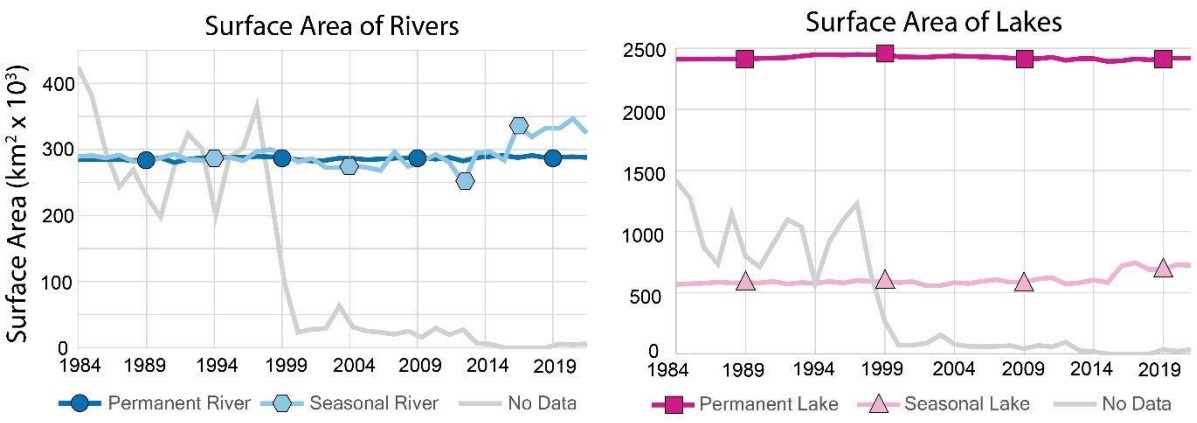


Figure 2: Global Water Surface Area Summary - Graphs show the temporal variability in permanent and seasonal water extent
of rivers and lakes by year.

The spatial trends in the total water surface area change since the first observation until the year 2022 are often
similar for rivers and lakes. Here we see that regions of the Basin and Range in the United States, southern South
America and Patagonia, southern Africa, Central Asia and Central Australia, show a decrease in water body
extent. In contrast, significant regions around the equator show increased waterbody extent including Brazil,
central Africa and oceania. In addition, the northern latitude Canadian Shield and Siberian plateau, as well as the
Himalayas, Europe, China, Southeast Asia and India show increasing water trends to name a few. Lastly, no
water observations are most commonly found in desert regions of Northern African Sahara, Southwestern Africa,
Western Australia and the Arabian peninsula, as well as glacial covered northern regions of Nunavut and
Greenland. It is crucial to emphasize that Figure 3 presents a comparison between only two time periods, and
may be skewed by variation over time with a shorter interval than the two time periods. We address this point
later in section 3.2 through spearman correlations analyses of water surface area trends over time.

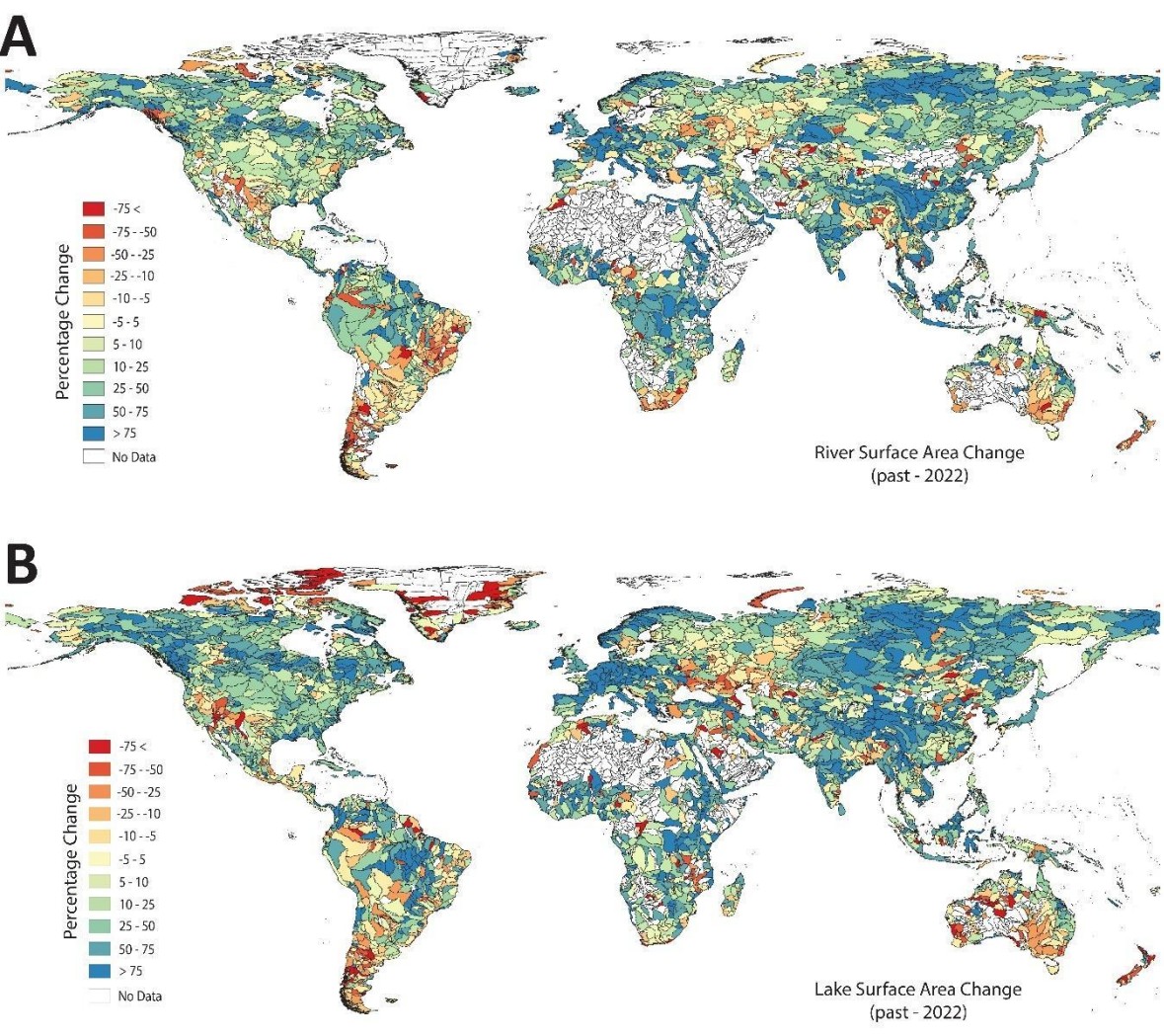



Figure 3: Global Total Water Surface Area Change - The global total surface water area of permanent and seasonal extent of
rivers (A) and lakes (B) based on the difference between the first recorded observation to 2022. See supplementary Figure S3
for percentage change by permanent and seasonal river and lake levels.

**3.2 Annual Water Surface Area Trends**

Figure 4 shows trends in permanent versus seasonal water surface area of rivers and lakes from 1984 to 2022 based
on the Spearman rank correlations (rho). Permanent river and lake extents show a relative normal distribution in
spearman correlations indicating the permanent extent of rivers and lakes have experienced both decrease and
increase in surface water area. In total, 47% and 54% of catchments show the permanent surface area of rivers
(Figure 4A) and lakes (Figure 4C) have statistically changed. Out of all catchments with statistically significant
changes, 62% and 55% of catchments have a positive increase in permanent water surface area for rivers and lakes,
respectively. In comparison, the seasonal extent of rivers and lakes is strongly positive indicating an overall
increased seasonality with 42% and 49% of catchments showing a significant change for rivers and lakes,
respectively (Figure 4B and 4D). Out of all catchments with statistically significant changes, 84% and 90% of
catchments have a positive increase in seasonal water surface area extent for rivers and lakes, respectively.



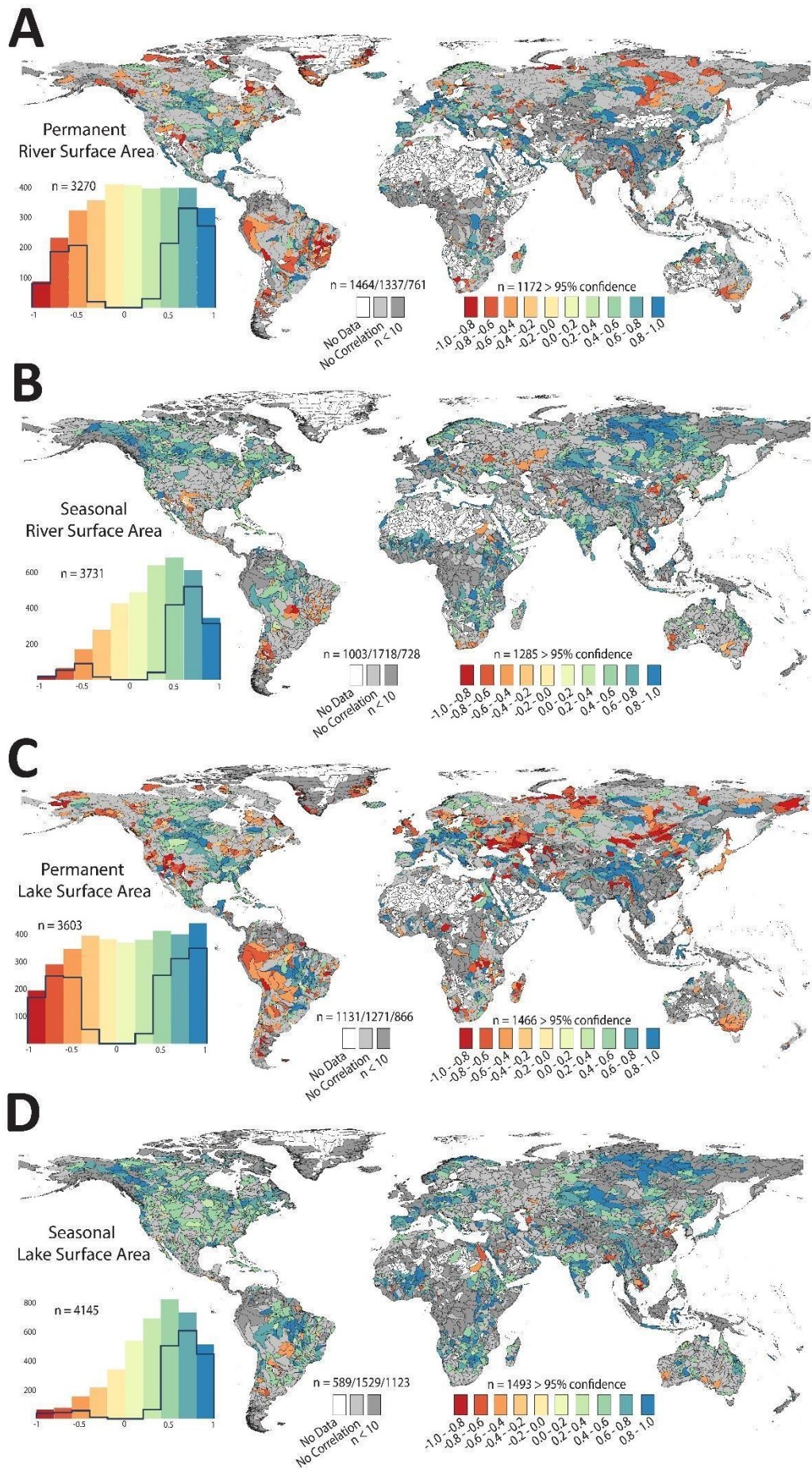

Figure 4: Spearman Rank Correlations: Catchments with statistically significant change in permanent (A,C) and seasonal
(B,D) water surface area since 1984 to 2020 for rivers and lakes. Spearman rank correlations (rho) are shown ranging from -1
(red) to 1 (blue) where no correlation (light gray) indicates no statistically significant change. The histogram for each map
shows the spearman correlations with the line indicating the distribution of the statistically significant samples (n > 9). See
supplementary Figure S4 for the entire spearman correlation dataset.

**3.3 Accuracy**

Overall, the SARL dataset showed a 93.8% accuracy to the manual delineation of the channel belt extents for the
calculation of permanent and seasonal lake coverage over the 38-year period as summarized in Figure 5. The
permanent extent of lakes is the most consistent and has the highest accuracy ranging between 96.7 and 98.5%.
The accuracy of the permanent river extent is lower and ranges from 94.3 to 97.5%. Finally, both the seasonal
river and seasonal lakes water bodies show on average an accuracy over 90% but are also the most variable
reflecting the variability in yearly water extent. The accuracy of the seasonal river is slightly higher ranging from
87 to 95% whereas seasonal lake has the lowest reported accuracy between 85 and 95%.

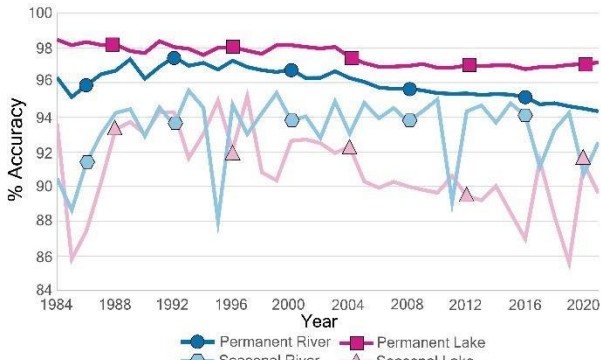

Figure 5: Temporal Accuracy of the SARL Database - Overall accuracy of the SARL database for seasonal and permanent
water extent of lakes and rivers.

**4. Discussion**

**4.1 Global Surface Water Trends**
Global observations of surface water extent are important to regulate and manage water at a basin scale for a range
of different sectors including agriculture, ecosystems, forestry, energy, water supply, environment protection,
flood and drought control, irrigation, wastewater treatment, governance and policy to name a few (Garcia et al.,
2016; Sheffield et al., 2018). In particular, the data is important to understand the impacts of hydrological extremes
of droughts and floods on the short and long-term trends on water allocation and management. The distinction of
rivers and lakes is important to understand the storage and transfer of water that impact different sectors and to
predict the future impacts of human and climate change stressors.
The permanent versus seasonal extent of water is also important to quantify pressures on water resources. In many
circumstances, the permanent extent of rivers and lakes has decreased over the past ~4 decades of observations
(38 and 45% of watersheds, respectively) and replaced by an increased seasonality (60 and 77%, respectively;
Figure 4, Pekel et al., 2016; Donchyts, et al., 2016). A change from perennial to seasonal surface water of rivers
directly impacts groundwater recharge and the resulting water table level (Gleeson et al., 2020). For instance,
regions of central Australia, Caspian Sea, Aral Sea, western United States and southern Africa have seen a
statistical decrease in permanent waters of both rivers and lakes (Figure 4) that are related to well documented
droughts and anthropogenic stresses (Micklin et al., 2007, Van Loon et al., 2016; Mekonnen et al., 2016). There
are also rivers that have experienced relatively stable permanent river water levels while also showing a decrease
in annual seasonal flooding, e.g., the Ob River in Siberia (Zemtsov, 2019).

On the other extreme, the Brahmaputra River in Bangladesh and India show stable permanent water levels with an
increased seasonal extent (Figures 2 and 4) that correlates to increased intensity of river discharge during
monsoonal seasons due to increasing Southern Oscillation Index extremes (Mirza, 2011). In other natural river
systems, like the lower Amazon basin, we see a slight, statistically significant, increase in water surface area of
rivers over past three decades (Figure 4). The Amazon water catchment as one of the least human-modified river
systems globally (Grill et al., 2019), has increased water levels due to a strengthening Walker circulation
(Barichivich et al., 2023). An increase in permanent water surface area for both rivers and lakes is particularly
noticeable in central Asia and the Tibetan Plateau as well, which has been associated with the acceleration of
glacial melting and precipitation (Bo Huang, 2014). Furthermore, seasonal water expansion in Siberia can be
related to increased thawing of permafrost lakes in summer months (Matthews et al., 2020). Lastly, reservoir
expansion, particularly around the Indian subcontinent, eastern Brazil and China, contribute to the increased water
surface area of lakes (Donchyts et al., 2022). In summary, statistically increasing permanent extent in rivers and
lakes account for 62% and 55% of watersheds, respectively (Figure 4), and often show a constant or increasing
seasonal extent as well (rho > -0.05; 78 and 88%, respectively), to suggest larger water bodies through time.


**4.2 Ecosystem Health**
Understanding temporal and spatial changes in rivers and lakes is key to the study of animal migrations and
community dynamics in and around lotic and lentic environments (Ngor et al., 2018). The availability of spawning
pathways from ocean to permanent mainstem rivers and lakes to permanent or seasonal tributary streams and lakes
is a function of water presence and quality (Briggs et al., 2018). Site selection by aquatic organisms of preferred
habitat for feeding and nutrients will also depend on whether or not water is present at candidate locations
(Power et al., 2008). In general, riverine and lacustrine animals move in response to changing water extents, so the
ability to determine or predict the occurrence of water in global rivers and lakes is a powerful capability.
The ability to couple the area of river and stream surface water with adjacent, in-channel terrestrial area supports
increased understanding of freshwater microhabitats and freshwater biotic interactions. For example, while fish
are obviously restricted to in-water lotic microhabitats like riffles, pools and runs (whose locations also change

spatially and temporally with changing surface water extent), amphibians and certain terrestrial invertebrates regularly move between water and adjacent in-channel dry land (Lowe, 2009). Conceptualizing and delineating freshwater ecosystems as only containing water is therefore under-representative of the area of occupancy and use by many freshwater aquatic organisms.

Certain areas in the aqueous stream channel are utilized as flow refugia (Lancaster and Hildrew, 1993; Sakai et al., 2021) or refugia from adverse stream acidification episodes (Baker et al., 1996) during discharge events. The availability of flow refugia is a function of riverine surface water seasonality (Lancaster and Hildrew, 1993). Clearly, the ability to bound the riverine environment as not just the water but the larger area within the channel belt and then to be able to distinguish between water and adjacent dryland spatially and temporally within that riverine ecosystem, will advance understanding of the distribution and behavior of aquatic organisms in seasonally changing environments. Moreover, current biodiversity models often overlook the importance of terrestrial and aquatic ecosystems in non-perennial systems (Messager et al., 2021). The current datasets may help to narrow this knowledge gap.

The delineation of channel belt areas as geomorphologically-derived riverine environments also has potential for addressing the 'linearity' challenge when delineating global freshwater ecosystems and habitats. For certain applications like ecosystem conservation status reporting (such as is required by the UN Convention on Biological Diversity) and ecosystem accounting (such as is characterized in the guidance from the UN's System for Environmental and Economic Accounting), area-based measures of ecosystem extent are needed. Except for very large rivers, however, river features are nearly always represented spatially as vector networks where the spatial entity for a river reach is a line segment. The segments may have an attribute for river width, but regardless, the spatial entities representing river reaches are generally not area-based and including freshwater ecosystems in area-based assessments is challenging. As such, there may be utility in using the Global Channel Belt resource for the spatial delineation of global freshwater ecosystems which would permit area-based assessments of their condition.

**4.3 Biogeochemical cycles**

Existing observations of river surface area at a 30 m global resolution suggest an area of $4.6 \times 10^5$ km at mean annual water level (Allen and Pavelsky, 2018). However, our current study at the same 30 m resolution suggests that the permanent extent of rivers is considerably lower at $2.9 \times 10^5$ km$^2$ or 37% less (Figure 2). On the other hand, the seasonal extent of rivers may contribute another $3.2 \times 10^5$ km$^2$ for a total area of $6.1 \times 10^5$ km$^2$ or an area approximately 32% larger than the previous observed estimate. Given rivers are known as a significant source of carbon emissions through water-atmosphere controls, our observations suggest a strong seasonal influence on biogeochemical cycles. Indeed, current estimates based on modelled monthly river surface area suggest that rivers emit $2.0 \pm 0.2$ Pg C y$^{-1}$ and that it is strongly based on seasonal river extents, particularly in temperate and arctic rivers (Liu et al., 2022). The current study furthermore suggests that rivers have, over the past 38 years, increased in seasonality by as much as 12%. This indicates that carbon emissions from rivers have significantly increased in recent years, especially from high latitude Arctic rivers and high mountainous regions, of for instance, the Tibetan Plateau that experience longer summer months.

The type of river system is also important in the carbon cycle with high discharge braided rivers recognized to
actively erode carbon rich floodplain material to reduce carbon oxidation into the atmosphere (Repasch et al.,
2021). The current SARL database show locations of laterally active river systems versus human-controlled river
systems. It is also important to recognize the perennial versus non-perennial nature of rivers and lakes which
suggests that $CO^2$ emissions from dry inland waters is overlooked in global calculations contributing an additional
6% (~0.12 Pg C $y^{-1}$; Keller et al., 2020; Messager et al., 2021). Because we found many rivers and lakes have
decreasing permanent extents (40 and 46%, respectively; Figure 4), our results suggest that previously buried
sediments with a disproportionately high amount of organic carbon will be increasingly released to the atmosphere
(Keller et al., 2020; Hao et al., 2021). Lastly, the damming of rivers and its impact on flow and sedimentation
patterns have been shown to eliminate another 48±11 Tg C $y^{-1}$ (Maavara et al., 2017), although there remain
significant knowledge gaps at the local to regional level. The current study and recent reservoir maps (Donchyts
et al., 2022; Khandelwal et al., 2022) highlight that rivers and reservoirs have undergone significant historical
changes, which may help to further constrain these estimates.

**5. Conclusions**

Here we have presented a new classification on the long-term change in permanent and seasonal extents of both
rivers and lakes from 1984 to 2022. Our results show that while the global area of permanent rivers and lakes has
remained relatively steady over the past 4 decades (~1% change), the regional variability is considerably higher.
The global extent of seasonal rivers and lakes present a different trend, increasing in surface area by 12% and 27%,
respectively. For catchments with a statistically significant change, 84% of rivers and lakes are positively skewed
showing an increased seasonality in surface water coverage over the same period. Decreasing perennial extent of
many rivers and lakes (38 to 45%, respectively) is often reflected in an increased seasonality of those same water
bodies (60 and 77%, respectively). However, an increasing perennial extent of rivers and lakes (62% and 55%,
respectively) also show a constant or increased seasonal coverage of the same water bodies (78 and 88%,
respectively) to create an overall expanding maximum surface area extent annually. Quantifying perennial and
seasonal change of rivers and lakes is crucial for measuring and tracking the health of aquatic ecosystems and the
impact of climate change and human pressures. The strong increase in seasonal maximum extent of rivers suggests
atmospheric-carbon interactions in rivers may have been larger than expected from permanent river coverage
alone, and may also have increased over the last few decades. The results of our analysis are shared as the SARL
database, which includes waterbody type, seasonality and spatio-temporal change for global rivers and lakes. This
database is a valuable resource and framework for water resource monitoring and assessment of ecosystem health
and conservation for different waterbody types.

**Data Availability**
The SARL database developed in this study has been deposited in the Zenodo database under accession code
https://doi.org/10.5281/zenodo.6895820.   An interactive map is available at
https://bjornburrnyberg.users.earthengine.app/view/waterchange

**Author Contribution**
BN conceived the original idea, designed the methodology and created the database. BN and EL analyzed the
original data and created the figures. BN, RS and EL wrote the manuscript.
**Competing interests**
The authors declare that they have no conflict of interest.

**Acknowledgements**
Any use of trade, product, or firm names is for descriptive purposes only and does not imply endorsement by the
U.S. Government. The authors are grateful for the journal-provided reviews and for helpful comments from John
W. Jones of the U.S. Geological Survey. Nyberg was funded by the Architectural Element Characterization of
Fluvial Systems project by AkerBP ASA and the Sea Level Projections and Reconstructions (SeaPR) project at
the Bjerknes Centre for Climate Research.

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
