# Peer review of "Increasing seasonal variation in the extent of rivers and lakes"

_EGUsphere, 2023_

## Referee Comment (RC1)

**General Comments:**

Surface water area plays an important role in ecosystems, the carbon cycle, flood and drought risks, and water resource management. Global surface water datasets built on moderate resolution satellite imagery (e.g., Landsat with 30-m, 16-day resolutions) have been developed, but they do not differentiate types of surface water bodies. The authors created a new dataset of the surface area of rivers and lakes (SARL) with seasonal and permanent surface water data for each year from 1984-2022. They developed this new dataset by combining data from the Global Surface Water dataset (Pekel et al., 2016) and from the global extent of river channel belts dataset (Nyberg et a., 2023). They were also able to assess changes from 1984 to 2022 in permanent and seasonal surface water areas in watersheds across the globe. They found seasonal surface water area had increased for both rivers and lakes, while the global total permanent surface water area was relatively unchanged. This dataset, showing where and when there are changes in lake and river permanent and seasonal surface water areas, can be helpful for water resource management. Such management practices are used to help limit the adverse impacts of extreme events such as floods and droughts, which are becoming more common due to climate change. The SARL dataset can help identify changes in the seasonality or permanence of surface water areas of key stormwater or drinking water reservoirs and help inform decision making about these water bodies in the future.

**Specific Comments:**

Figure 1 B: i) The caption suggest image B is showing the change in permanent water extent from 1984 to 2020; however, the legend in the image suggests it is the permanent and seasonal extents for one year, i.e., there is no legend entry to indicate a pixel has changed classification. Please change either the figure or the caption to clarify what you are showing. ii) it is hard to distinguish the Permanent Lake from the Seasonal Lake. Consider using a different color for one of them. iii) There is already green in the Landsat image, so it is hard to distinguish the green for Seasonal River. Consider showing the Landsat image using false color or changing the Seasonal River color to something easier to identify.

Line 111-113: How did you use the HydroLakes and OpenStreetMap datasets? I'm guessing you removed GCB pixels outside of the HydroLakes and OpenStreetMap bounds. Also, I'm assuming the data product of GCB is pixel-based rather than shapefiles, but I'm not clear on that.

Also, have you looked at what areas are being removed from these bounds? By looking at your GEE app, I found an example of portions of the James River outside of Richmond, Virginia in the United States being excluded from the watershed. The exclusion of such large areas of what I would assume is permanent river area may be impacting your results. I think a discussion of this limitation would be valuable to add.

[Figure]

Line 124: The GSW database does not identify waterbody type and the missing/no data values can be non-water, as well as permanent or seasonal surface water. How does your method account for this possibility?

Line 125-126: Because you are suggesting that seasonal water surface area has increased between 1984 and 2022, why do you think using the average ratio of seasonal to permanent waterbody extents between 2015 and 2017 is a good ratio to use across all of the years? (aside from those years lack of missing values.) I would suggest moving the sentence from 145-147 up to this paragraph.

Line 135-136/Equation 1: Please clarify the units of the values in the equations. It reads like pArea, pO, sArea, and sO are all measures of area and therefore the units are something like ha or $m^2$. However, the nD appears to be a count of no data values (or pixels) in the catchment. I expect you are not adding area and pixel counts together, to find pArea and sArea, but please clarify.

Line 171-172: because you are using the same surface water dataset to generate the validation dataset as you used to generate the SARL dataset (GSW), this validation assessment seems to only be validating the GCB dataset. If you want to validate the SARL dataset, you might want to consider classifying the Landsat 8 images you used to manually delineate channel belts into permanent, seasonal and non-water classes; use the manually delineate channel belts to classify the permanent and seasonal water into riverine and lacustrine; and then compare these validation data to the SARL data.

Line 169-176: This is the only paragraph where you refer to the SARL dataset as being generated through an "automated method". What I assume this means, is that the GCB data used to create the SARL dataset is produced using an automated method, as compared to the manual delineation of the channel belt used to generate the validation dataset. The final sentence of the paragraph supports my assumption that this validation analysis is focused on validating how well the GCB dataset performs, rather than validating the SARL dataset. This is fine, but if that is the goal, consider restructuring the aim of the paper from "to compile existing information to map the historical change in water surface area…" to "to assess the utility of a new, global river extent dataset (Nyberg et al., 2023) in mapping the historical change in water surface area…"

Line 222: What are the "independent estimates of permanent and seasonal lake coverage" to which you are comparing the SARL dataset?

**Technical Corrections:**

Line 43: typo -- "seasonall" to "seasonal"

Line 46: "their biodiversity" whose biodiversity? The biodiversity of human or ecosystems or something else?

Line 46: use of an Oxford comma where in the previous sentence one is not used. Recommend picking one (use it or don't use it) and make it consistent throughout the manuscript

Line 63: typo/subject-verb agreement: "provides" to "provide"

Line 78: If you are getting the GSW data from GEE, 1) which GSW data product are you using? I assume you are using the JRC Yearly Water Classification History, v1.4 ("JRC/GSW1_4/YearlyHistory"). 2) if that is the case, that product only runs through 2022, not 2023. Please include which version of the GSW product you are using and how you accessed it.

Line 91: did you evaluate the model in this current study, or is the evaluation from Nyberg et al., 2023? If the latter, consider rephrasing to "The model reported … (Nyberg et al., 2023)." or "The authors reported the model had a 94% accuracy …"

Line 93-94: consider rephrasing "surface area of lakes and rivers (SARL)" to "surface area of and rivers lakes (SARL)"

Line 94: is the SARL dataset from 1984-2022 (as in the title) or 2023?

Line 96: consider rephrasing the last sentence of this paragraph. It sounds like the beginning and end of the sentence are saying the same thing. You could probably end the sentence after "wetland regions", unless I'm not understanding what you're trying to say.

Figure 1/Line 100: is the GCB a model or a dataset produced by a model?

Line 106: rephrase for methods clarity. From "classification of Nyberg et al., (2023) by utilizing a 10% confidence on the GCB prediction… and a 50% confidence…" to "classification of Nyberg et al. (2023) by using GCB pixels with a reported confidence of 10% or higher…and a 50% or higher confidence…" or "classification of Nyberg et al. (2023) by utilizing GCB pixels with a ≥10% confidence or higher… and a ≥50% confidence…".

Line 111-113: use of passive voice ("datasets were included" and "these steps were processed") after using active voice in the preceding paragraphs. I recommend you pick one and stick with it throughout the manuscript.

Section 2.1: I suggest a slight restructuring of this section to clarify the steps taken to generate the SARL dataset. Currently, it reads as:

     a) Describe GSW (Pekel et al., 2016)
     b) Describe GCB (Nyberg et al., 2023)
         a. Discuss combing GSW and GCB
     c) Describe how GCB was subset using confidence thresholds and other lake/reservoir datasets

I suggest reordering it so it flows more like:

a) Describe GSW (Pekel et al., 2016)
b) Describe GCB (Nyberg et al., 2023)
c) Describe how GCB was subset using confidence thresholds and other lake/reservoir datasets
d) Describe how GSW and the subset GCB were combined to create SARL

Line 153: How many HydroSHEDS catchments are there globally versus how many did you all use in your analyses? Phrased another way, how many catchments are you removing with the qualifications stated in this sentence?

Line 174: You suggest that you're creating this validation dataset using GSW data from 1984-2023, but you are only using the data from 2022 for validation (see lines 160-161). Consider rephrasing this sentence to clarify how much of the GSW data you are using for validation.

Line 224-225/Figure 4: Figure 4 shows Permanent River accuracy is between 94-98% and the second highest accuracy range, but your sentence says it has the lowest accuracy range (84-91%). Please check which is correct and update.

Line 262: suggest removing "e.g."

Line 273: I'm unclear on how area and mean annual water discharge are linked here. Consider rephrasing for clarity.

Line 282: suggest changing "entails" to "indicates"

Section 4.3: This section is titled "Water Resource Management" but I am not seeing much discussion about how the SARL dataset will be used for water resource management. The paragraphs have lots of good information, but it would be nice to have the connection to the SARL dataset spelled out a little more explicitly.

Line 310: change "river" to "rivers"

Line 318: Change "show" to "shows"; delete "the" before "seasonal extent"

Line 319: Change "is attributed" to "correlates"

Line 321: Add comma between "significant" and "increase"

Line 325: Add comma between "well" and "which"

Line 338: Are you talking about catchments with any kind of statistically significant change (i.e., permanent lake or seasonal river) or are you talking about catchments with a specific kind of statistically significant change? Please clarify.

**Minor rephrasing suggestions**

These comments are just suggestions of rephrasing for clarity and do not need to be addressed.

Line 80: consider rephrasing "…with at least one month during which water was detected" to "where water was detected for at least one month" (optional, it's okay as-is but rephrasing might make it a little clearer)

Line 85-87: consider rephrasing "Nyberg et al. (2023) quantified…" to "We used a dataset that quantified the global extent of river channel belts (GCBs) … produced by Nyberg et al. (2023). In this dataset, the river channel belt…"

Line 89: consider rephrasing "The model reports" to "The model used to produce this dataset reports…" or "The model used by Nyberg et al. (2023) reports…"

Line 159: suggest rephrasing "available water surface change data" to "available surface water data"

Line 169: suggest removing the term "proposed". You have utilized the method, so I do not think you are proposing it. I would rephrase the sentence to "We used the same method to classify the validation dataset as we used to classify the SARL dataset (Figure 1), where waterbodies defined…"

Line 291: I suggest rephrasing "The decreasing permanent extent of many lakes and rivers (Figure 2) suggests that…" to "Because we found many (XX and XX%) lakes and rivers have decreasing permanent extents (Figure 2), our results suggest that…" Right now, the phrasing sounds like the majority of the lakes and rivers are decreasing in permanent area. Whereas in Figure 3, it looks like there is a somewhat even split between statistically significant increasing and decreasing permanent water area for lakes and rivers, with somewhat more showing increases.

Line 308-309: Same idea as above; the use of the term "many" seems to imply that at least close to the majority if not the majority of permanent water in lakes and rivers is decreasing.

Line 314-316: consider rephrasing to "There are also rivers that have experienced relatively stable permanent river water levels while also showing a decrease in annual seasonal flooding, e.g., the Ob River in Siberia (Zemtsov, 2019)."

---

## Author Comment (AC1)

Reviewer #1

**Specific Comments:**

Figure 1 B: i) The caption suggest image B is showing the change in permanent water extent from 1984 to 2020; however, the legend in the image suggests it is the permanent and seasonal extents for one year, i.e., there is no legend entry to indicate a pixel has changed classification. Please change either the figure or the caption to clarify what you are showing. ii) it is hard to distinguish the Permanent Lake from the Seasonal Lake. Consider using a different color for one of them. iii) There is already green in the Landsat image, so it is hard to distinguish the green for Seasonal River. Consider showing the Landsat image using false color or changing the Seasonal River color to something easier to identify.

*A – Correct, we have changed the figure caption in the updated manuscript to be clear that Figure 1B shows one year of permanent and seasonal extents. The original figure caption meant to indicate that permanent and seasonal extents do exist for every year from 1984 to 2022. In addition, we have taken the reviewers advice to change the color scheme for both seasonal lake and seasonal river extents to a more colorblind friendly scale.*

Line 111-113: How did you use the HydroLakes and OpenStreetMap datasets? I'm guessing you removed GCB pixels outside of the HydroLakes and OpenStreetMap bounds. Also, I'm assuming the data product of GCB is pixel-based rather than shapefiles, but I'm not clear on that.

Also, have you looked at what areas are being removed from these bounds? By looking at your GEE app, I found an example of portions of the James River outside of Richmond, Virginia in the United States being excluded from the watershed. The exclusion of such large areas of what I would assume is permanent river area may be impacting your results. I think a discussion of this limitation would be valuable to add.

*A – Thank you for that excellent point and we have now addressed this issue by adding a couple sentences to the methodology to indicate that the HydroLakes and OpenStreetMap datasets are converted to rasters first. Once all datasets are rasters, the lacustrine area from those combined datasets can be removed from the GCB delineation extent to include all the available geospatial information.*

*The example of the James River in Virginia excluding the classification is due to the shoreline delineation provided by the global shoreline vector (GSV) dataset which is used in the current study and the GCB dataset. The shoreline is needed to classify pixels that are oceanic and should therefore not be considered in the SARL database. This approach does have limitations as the definition of the coastline may vary (e.g. high tide, MHHW level or river thalweg distance). We have taken the reviewers suggestion to add a new paragraph to the methods sections to discuss this seaward extent and limitation.*

Line 124: The GSW database does not identify waterbody type and the missing/no data values can be non-water, as well as permanent or seasonal surface water. How does your method account for this possibility?

*A - Here we account for this shortcoming by only considering those years that have at least 95% data coverage within each watershed catchment. We assume that the remaining 5% of the datasets are water and take the ratio of the seasonal to permanent to calculate the estimated distribution. Similarly, the ratio between lake, river and no water is used to calculate the waterbody type for the missing 5% of the pixels. This step was chosen to have a more comprehensive dataset when only a few pixels are missing, likely due to cloud coverage.*

Line 125-126: Because you are suggesting that seasonal water surface area has increased between 1984 and 2022, why do you think using the average ratio of seasonal to permanent waterbody extents between 2015 and 2017 is a good ratio to use across all of the years? (aside from those years lack of missing values.) I would suggest moving the sentence from 145-147 up to this paragraph.

*A -  Our preliminary results suggest there were circumstances where a interpolation may not work, in particular, the seasonal extent is often missing in earlier GSW acquisitions (i.e. predominately permanent classifications) leading to omitted seasonal extent. Given we are only examining catchments with less than 5% missing values, we assume that the 2015-2017 ratio will not significantly alter the overall trend while adding a more complete dataset. We have clarified and emphasized this point in the updated manuscript.*

Line 135-136/Equation 1: Please clarify the units of the values in the equations. It reads like pArea, pO, sArea, and sO are all measures of area and therefore the units are something like ha or $m^2$. However, the nD appears to be a count of no data values (or pixels) in the catchment. I expect you are not adding area and pixel counts together, to find pArea and sArea, but please clarify.

*A – This section has now been clarified. Given each pixel has an area (width x height), the number of pixels also refers to a surface area. We have changed the mention of nD in terms of area to be consistent with the other variables.*

Line 171-172: because you are using the same surface water dataset to generate the validation dataset as you used to generate the SARL dataset (GSW), this validation assessment seems to only be validating the GCB dataset. If you want to validate the SARL dataset, you might want to consider classifying the Landsat 8 images you used to manually delineate channel belts into permanent, seasonal and non-water classes; use the manually delineate channel belts to classify the permanent and seasonal water into riverine and lacustrine; and then compare these validation data to the SARL data.

*A – Indeed, the aim of the accuracy section is to validate the impact of the GCB delineation on the calculation of permanent and seasonal extent of rivers and lakes using the GSW dataset. While we agree that a manual classification of each channel belt into permanent, seasonal and non-water classes would be ideal, this would require each month over the 38-year period to be manually mapped for each location. This would require 22,800 scenes to be manually mapped (38 years x 12 months x 50 locations), which was deemed unfeasible within the timeframe of the project. Hence, we assume that the GSW classification which reports a >95% accuracy is correct and use that as a basis to validate the SARL database.*

Line 169-176: This is the only paragraph where you refer to the SARL dataset as being generated through
an "automated method". What I assume this means, is that the GCB data used to create the SARL dataset is produced using an automated method, as compared to the manual delineation of the channel belt used to generate the validation dataset. The final sentence of the paragraph supports my assumption that this validation analysis is focused on validating how well the GCB dataset performs, rather than validating the SARL dataset. This is fine, but if that is the goal, consider restructuring the aim of the paper from "to compile existing information to map the historical change in water surface area…" to "to assess the utility of a new, global river extent dataset (Nyberg et al., 2023) in mapping the historical change in water surface area…"

*A – This is correct. By combining the GCB dataset with the GSW, the SARL database is generated through an automated process. We agree that the phrasing should be clearer, and we have incorporated the reviewers suggested wording in the revised manuscript.*

Line 222: What are the "independent estimates of permanent and seasonal lake coverage" to which you are comparing the SARL dataset?

*A – This wording is incorrect and refers to the manually delineated channel belt used to define the manually delineated permanent and seasonal lake extents based on the GSW dataset as described in the validation section. We have corrected this by rephrasing to "permanent and seasonal river and lake coverage based on the manually defined channel belt delineations".*

**Technical Corrections:**

Line 43: typo -- "seasonall" to "seasonal"

*A - Changed*

Line 46: "their biodiversity" whose biodiversity? The biodiversity of human or ecosystems or something else?

*A - This refers to the biodiversity of the ecosystems. Changed to "alters migration patterns of humans, ecosystems, and biodiversity".*

Line 46: use of an Oxford comma where in the previous sentence one is not used. Recommend picking one (use it or don't use it) and make it consistent throughout the manuscript

*A – Changed. Have checked the document for consistency.*

Line 63: typo/subject-verb agreement: "provides" to "provide"

*A - Fixed*

Line 78: If you are getting the GSW data from GEE, 1) which GSW data product are you using? I assume you are using the JRC Yearly Water Classification History, v1.4 ("JRC/GSW1_4/YearlyHistory"). 2) if that is the case, that product only runs through 2022, not 2023. Please include which version of the GSW product you are using and how you accessed it.

*A – Yes the product is the GSW version 1.4 that runs 1984 – 2022. This has been corrected.*

Line 91: did you evaluate the model in this current study, or is the evaluation from Nyberg et al., 2023? If the latter, consider rephrasing to "The model reported … (Nyberg et al., 2023)." or "The authors reported the model had a 94% accuracy …"

*A – Fixed.*

Line 93-94: consider rephrasing "surface area of lakes and rivers (SARL)" to "surface area of and rivers lakes (SARL)"

*A - Changed*

Line 94: is the SARL dataset from 1984-2022 (as in the title) or 2023?

*A – Yes the current dataset is from 1984-2022 based on the available GSW extent from 1984 to 2022.*

Line 96: consider rephrasing the last sentence of this paragraph. It sounds like the beginning and end of the sentence are saying the same thing. You could probably end the sentence after "wetland regions", unless I'm not understanding what you're trying to say.

*A – Fixed*

Figure 1/Line 100: is the GCB a model or a dataset produced by a model?

*A – The GCB is a dataset produced by a model. This has now been clarified.*

Line 106: rephrase for methods clarity. From "classification of Nyberg et al., (2023) by utilizing a 10% confidence on the GCB prediction… and a 50% confidence…" to "classification of Nyberg et al. (2023) by using GCB pixels with a reported confidence of 10% or higher…and a 50% or higher confidence…" or "classification of Nyberg et al. (2023) by utilizing GCB pixels with a ≥10% confidence or higher… and a ≥50% confidence…".

*A - This is a good clarification and we have implemented the suggestion.*

Line 111-113: use of passive voice ("datasets were included" and "these steps were processed") after using active voice in the preceding paragraphs. I recommend you pick one and stick with it throughout the manuscript.

*A – We have checked the manuscript and opted for an active voice throughout.*

Section 2.1: I suggest a slight restructuring of this section to clarify the steps taken to generate the SARL dataset. Currently, it reads as:

a) Describe GSW (Pekel et al., 2016)
b) Describe GCB (Nyberg et al., 2023)
  a. Discuss combing GSW and GCB
c) Describe how GCB was subset using confidence thresholds and other lake/reservoir datasets

I suggest reordering it so it flows more like:

a) Describe GSW (Pekel et al., 2016)
b) Describe GCB (Nyberg et al., 2023)
c) Describe how GCB was subset using confidence thresholds and other lake/reservoir datasets
d) Describe how GSW and the subset GCB were combined to create SARL

*A – Thanks for the suggestion. We have rearranged the paragraph slightly to discuss the GCB preprocessing steps first before its combination with the GSW dataset.*

Line 153: How many HydroSHEDS catchments are there globally versus how many did you all use in your analyses? Phrased another way, how many catchments are you removing with the qualifications stated in this sentence?

*A – The numbers for each seasonal and permanent category for rivers and lakes are shown in Figure 3. However, we have now added this text in the paragraph for additional clarity.*

Line 174: You suggest that you're creating this validation dataset using GSW data from 1984-2023, but you are only using the data from 2022 for validation (see lines 160-161). Consider rephrasing this sentence to clarify how much of the GSW data you are using for validation.

*A – This sentence has now been rephrased to indicate that the channel belt validation is only from one year to assess the impact on the 38-years of available water detection.*

Line 224-225/Figure 4: Figure 4 shows Permanent River accuracy is between 94-98% and the second highest accuracy range, but your sentence says it has the lowest accuracy range (84-91%). Please check which is correct and update.

*A – This section has been double checked and updated with the correct numbers.*

Line 262: suggest removing "e.g."

*A - Done*

Line 273: I'm unclear on how area and mean annual water discharge are linked here. Consider rephrasing for clarity.

*A – This sentence has been rephrased for clarity. Surface area of rivers will change based on water levels throughout the year, which relates to water discharge.*

Line 282: suggest changing "entails" to "indicates"

*A – Changed*

Section 4.3: This section is titled "Water Resource Management" but I am not seeing much discussion about how the SARL dataset will be used for water resource management. The paragraphs have lots of good information, but it would be nice to have the connection to the SARL dataset spelled out a little more explicitly.

*A – We have added information in the introductory paragraph of the discussion to explicitly state how the SARL dataset may help within a water resource management perspective.*

Line 310: change "river" to "rivers"

*A - Changed*

Line 318: Change "show" to "shows"; delete "the" before "seasonal extent"

*A - Changed*

Line 319: Change "is attributed" to "correlates"

*A - Changed*

Line 321: Add comma between "significant" and "increase"

*A - Done*

Line 325: Add comma between "well" and "which"

*A - Done*

Line 338: Are you talking about catchments with any kind of statistically significant change (i.e., permanent lake or seasonal river) or are you talking about catchments with a specific kind of statistically significant change? Please clarify.

*A – This refers to both a all catchments with a statistically significant spearman correlation change (positive or negative). This sentence has now been clarified.*

**Minor rephrasing suggestions**

These comments are just suggestions of rephrasing for clarity and do not need to be addressed. Line 80: consider rephrasing "…with at least one month during which water was detected" to "where water was detected for at least one month" (optional, it's okay as-is but rephrasing might make it a little clearer)

Line 85-87: consider rephrasing "Nyberg et al. (2023) quantified…" to "We used a dataset that quantified the global extent of river channel belts (GCBs) … produced by Nyberg et al. (2023). In this dataset, the river channel belt…"

*A - Changed*

Line 89: consider rephrasing "The model reports" to "The model used to produce this dataset reports…" or "The model used by Nyberg et al. (2023) reports…"

*A - Changed*

Line 159: suggest rephrasing "available water surface change data" to "available surface water data"
Line 169: suggest removing the term "proposed". You have utilized the method, so I do not think you are proposing it. I would rephrase the sentence to "We used the same method to classify the validation dataset as we used to classify the SARL dataset (Figure 1), where waterbodies defined…"

*A - Changed*

Line 291: I suggest rephrasing "The decreasing permanent extent of many lakes and rivers (Figure 2) suggests that…" to "Because we found many (XX and XX%) lakes and rivers have decreasing permanent extents (Figure 2), our results suggest that…" Right now, the phrasing sounds like the majority of the lakes and rivers are decreasing in permanent area. Whereas in Figure 3, it looks like there is a somewhat even split between statistically significant increasing and decreasing permanent water area for lakes and rivers, with somewhat more showing increases.

*A - Implemented*

Line 308-309: Same idea as above; the use of the term "many" seems to imply that at least close to the majority if not the majority of permanent water in lakes and rivers is decreasing.

*A – This has been changed*

Line 314-316: consider rephrasing to "There are also rivers that have experienced relatively stable permanent river water levels while also showing a decrease in annual seasonal flooding, e.g., the Ob River in Siberia (Zemtsov, 2019)."

*A - Implemented*

---

## Author Comment (AC2)

Reviewer #2

**Major Comments:**

Q - Line 124: Can you clarify why you chose to average 2015 -2017 and use those values where you had missing data? Why did you opt not to interpolate or fill with the long term average or another metric?

*A – The 2015-2017 average was used to define the ratio between seasonal and permanent extent as it represents the GSW dataset years without missing datapoints thereby providing a complete global dataset of the ratio. While a long-term average of each pixel location may provide another solution, our approach provided a simple and efficient estimate to establish a complete dataset. Given that this was done for catchments with less than 5% of its pixels missing, we do not believe that this will significantly alter our findings. We have added additional text regarding this limitation in the updated manuscript. See also comments to reviewer #1.*

Results: It would be nice to include a general comment about regional differences or perhaps even climatic differences. Most importantly, are there regions that stick out or areas that are notably with respect to your results? I know this gets brought up in the discussion, but it might increase the clarity of your results.

*A – While it is beyond the scope of this paper to explore the climatic influences on the results, we have included a few mentions in the results to give more context. This section now ties better with the discussion later.*

Section 4.3: I would suggest changing this section title to something more specific. This section dives into the trends in and potential reasons for why the trends are occurring, but I don't directly see the link to water management strategies. I also think this would be a better first discussion paragraph as it goes into the reasons behind your results and would prepare the reader for the other two sections on ecosystem health and biogeochemical cycles.

*A – This is a valid point and we have now changed the paragraph order by moving section 4.3 to the start of the discussion. We have also changed the section title to "Global Surface Water Trends" which better describes the content of this section.*

Line 336: You discuss regional differences, but I don't feel like you dove into the regional differences as much. I would expand this in section 4.3. I would also elaborate on the regional differences you see in the conclusion.

*A – We have now expanded section 4.3 to include more discussion on the regional difference as well as including a summary of those regional differences in the conclusions.*

Conclusion: I would include one or two key points from section 4.1 and 4.2. This would make the conclusion a bit stronger and prove why this dataset is so unique and what it can be used for.

*A – This is a good suggestion and we have implemented the reviewers feedback in the conclusion by adding one sentence each on the biogeochemical implications and ecosystem health.*

**Minor comments:**

Line 31: The authors cite GRanD (Lehner et al., 2016), but it also might be useful to cite GeoDAR (Wang et al., 2023) as that is a more representative of the total storage that exists globally inside reservoirs.

*A - Added*

Figure 2: I would suggest breaking this up into two figures either by rivers and lakes or by plot type since the figure is quite tiny and it's hard to see the changes in the maps. I would also suggest making the lines on the line plots thicker and changing the colors to be more divergent, especially for the surface area of lakes (Panel B line plot). For the maps, I would include a color bar that shows that white denotes either no data or not enough data (I'm personally not sure which it is). Lastly, it could be beneficial to switch the color bar from rainbow to another one that is more color blind friendly.

*A – We have added a colorbar for the nodata symbol (not enough data is shown in Figure 3), increased the line thickness, and split the figure into two as the reviewer suggested. We have at this time decided not to switch the colorbar but can implement those suggestions as well if desired by the editorial team.*

Figure 3: I also suggest splitting this figure into two. Perhaps either by the type or by seasonal and permanent as I think it will increase the clarity of your results section. I also don't know if you need the color legend for each panel, but perhaps that is necessary?

*A – We have decided not to split the figure into two here considering section 3.2 contains only one paragraph that explains the figure. However, we can accommodate those suggestions if needed.*

Figure 4: I like this figure; however, I would change the colors to a different set (perhaps one color for each water body type and different shapes for seasonal and permanent). Seasonal Lake gets lost in this figure and my eyes are drawn to permanent lake and seasonal river.

*A – We have now changed the color scheme and included different shapes for seasonal and permanent waterbody accuracy for additional clarity.*